# Review of Fluorescence Spectroscopy in Environmental Quality Applications

**DOI:** 10.3390/molecules27154801

**Published:** 2022-07-27

**Authors:** Despoina-Eleni Zacharioudaki, Ioannis Fitilis, Melina Kotti

**Affiliations:** Department of Electronic Engineering, Hellenic Mediterranean University, Romanou 3, 73133 Chania, Greece; despoinazax@gmail.com (D.-E.Z.); fitilis@hmu.gr (I.F.)

**Keywords:** conventional lamps, fluorescence, laser sources, laser induced fluorescence, environmental quality

## Abstract

Fluorescence spectroscopy is an optical spectroscopic method that has been applied for the assessment of environmental quality extensively during the last 20 years. Most of the earlier works have used conventional light sources in spectrofluorometers to assess quality. Many recent works have used laser sources of light for the same purpose. The improvement of the energy sources and of the higher resolution spectrometers has led to a tremendous increase in applications. The motivation for the present review study is the increasing use of laser sources in environmental applications. The review is divided in two parts. The fundamental principles of fluorescence spectroscopy are described in the first part. The environmental applications are described in the second part.

## 1. Introduction

Optical spectroscopy deals with the study of interactions between matter and light such as absorption, emission and scattering, among others. Fluorescence spectroscopy (or fluorometry) is based on the emission of photons from a substance after excitation from light absorption. The molecules, due to their vibrational energy levels, emit light of lower energy (longer wavelength) than the absorbed light. This is called Stokes’ shift and happens due to an energy loss in non-radiative decay. These processes are shown in the so-called Jablonski diagram in Figure 1a. When a molecule is in the ground level (S_0_) and absorbs a photon of sufficient energy, an electron is promoted to a higher energy singlet level (S_1_ or S_2_) equal to the energy of the absorbed photon. If the electron, following a relaxation pathway, returns to its original state emitting a photon, it is called photoluminescence. The two types of photoluminescence are fluorescence and phosphorescence. Fluorescence refers to when the electron from a singlet excited level decays radiatively to the ground singlet state, within a characteristic decay time of the order of 10^−10^ s to 10^−7^ s. Phosphorescence occurs when the electron, after a transition from a singlet excited state to a triplet state, returns radiatively to the ground singlet state. Since the last process involves change of the electron spin, the phosphorescence time could be from 10^−6^ s up to seconds.

For a compound, its emission spectrum and absorption spectrum (excitation spectrum) are usually almost mirror images as shown in Figure 1b for the radiative transition from S_0_ to S_1_ and vice versa. This symmetry is attributed to the same vibrational levels structures that are involved in each of these processes. 

### 1.1. Light Sources

Fluorescence excitation can be induced by a lamp, a light-emitting diode (LED) or a laser source. When a laser is used for the excitation, the fluorescence is called laser induced fluorescence (LIF). As lamps are conventional sources of light, the fluorescence is considered as conventional in that case. For multispectral light sources, a spectral filter or monochromator is used for selecting the excitation wavelength. 

Typically, the commercial spectrofluorometers utilize arc-lamps as light sources for the excitation of compounds. The main sources used for ultraviolet (UV) and visible light are high-pressure and low-pressure lamps of Xenon (Xe) and mercury (Hg). The line spectrum emission of low-pressure Hg-Ar and Hg lamps are used mainly for calibration purposes. The high-pressure Xe and Hg lamps provide broadband light emissions from 250 nm and from 350 nm, respectively, up to IR. The intensities of high-pressure Hg lamps are higher compared with Xe lamps. There are also high-pressure Xe-Hg lamps with higher intensities than Xe lamps. Deuterium lamps are also used, providing continuum emission in UV, from 160 nm to 400 nm. The tungsten-halogen lamps are less used in fluorescence spectroscopy than the previously mentioned ones due to their weak light emission below 400 nm [2]. 

In the last two decades, the light-emitting diodes (LEDs) began to appear as light sources in spectrofluorometers. The LED is an electroluminencence device that produces photon emission by the recombination of electrons with electron holes at semiconductor junction. They are inexpensive light sources with bright illumination and the development of LEDs emitted up to deep UV range make them very attractive for fluorescence excitation [3].

The laser sources are categorized in four types according to the active medium: solid state, gas, liquid and semiconductors/diode lasers. Among the solid state lasers, the Neodynium-doped Yttrium/Aluminium Garnet (Nd: YAG) is the most common type used in fluorescence spectroscopy [4]. The fundamental wavelength of the laser is in the infrared (IR) at 1064 nm but with the use of nonlinear crystal the second or higher harmonics could be produced, providing laser wavelength at 532 nm, 355 nm, 266 nm or even 213 nm.

The most common gas lasers include argon ion, HeNe, nitrogen and excimers [5]. Argon ion lasers emit radiation at 488 nm (blue) and 514 nm (green); hence, they are called also blue-green lasers. They are relatively large lasers and external cooling is required. Helium-neon (HeNe) lasers operate at a single wavelength of light, most often at 632 nm. They do not produce high-power light (from few to tens mW), but they are more stable and are often used for metrology applications. HeNe lasers are more compact than argon ion lasers and do not require external cooling. Another type of gas lasers used in spectroscopy are nitrogen lasers, which operate in the UV range, as they emit radiation at 337 nm. They have been used for air pollution monitoring. Excimer lasers are another type of gas lasers for UV emission. The laser medium are short-live dimeric molecules generated in excited state of inner nobble gas atoms themselves (excimer) such as argon, krypton, xenon or by their combination with halogens (exciplex) such as fluorine or chlorine gas atoms. 

The liquid lasers consist of a dye as lasing material. An organic dye dissolved in water or other solvent (at a typical concentration of the order of 1 part in 10000), is radiated by an intense light source [6]. After the absorption of light in one wavelength, the dye emits light over a broad range of visible wavelengths. Different dyes can produce a variety of wavelength emissions. Coumarin and Rhodamine are some of the most used dyes [7,8]. By changing the cavity length, or through other techniques, the output wavelength can be easily tuned in a wide range of tens of nm. Their drawbacks are that the dye undergoes photodecomposition and needs to be renewed by circulation, either in a container cell or as stream of jet in open air. In addition, the dyes are hazardous toxic materials and great attention when handling them is needed. 

The last category of lasers is the diode (or semiconductor) lasers that work on a somewhat different principle. Light is emitted by flowing electrical current through the semiconductor due to the energy gap in the diode’s junction. They are compact lasers that emit wavelengths typically from visible to IR, but recently there are devices emitting in the UV region [9]. Their output has a wide range of power from low to moderate. They have small size, are lightweight and less expensive than other lasers [10]. 

Generally, the choice of laser type depends on the wavelength range that is needed in each research. The application of laser depends on the properties (monochromaticity, directionality, spatial and temporal characteristics) of the laser beam. 

### 1.2. Detectors

The fluorescence light is detected and quantified either by a photomultiplier tube (PMT) or avalanche photodiodes (APD) or by a charge-coupled device (CCD). The majority of commercial fluorometers use photomultiplier tubes (PMT) to detect the low light intensity of fluorescence [2]. A PMT is a vacuum glass tube that consists of a photocathode where incident photon induces electron emission, a series of dynodes for electron multiplication and an anode. The material that the photocathode is made from determines the spectral range of the photomultiplier. In order to cover the whole UV-visible range, two types of photocathodes are required. 

The APD is another low-intensity light detector used in fluorescence spectroscopy. In a photodiode an electron-hole pair generated when a photon is captured in the diode junction area. In APD, a high reverse bias voltage creates a strong electric field where the electron generated from photon is accelerated to produce secondary electrons by impact ionization. The resulting electron avalanche produces measurable electrical signal even from few photons [11].

Another type of detector used in fluorometers is the CCD with which the whole fluorescence spectrum can be analyzed at once without the need for wavelength scanning with a monochromator. It is mainly used when the emitted light is of higher intensity, such as in LIF. The CCD consists of an array of semiconductive photosensitive elements (pixels) where photons induce an electrical charge which is accumulatively stored in each of them. Then, the read out of each element charge is made in series by shifting the charge of each element to their neighbor toward the charge measurement circuit where it is also digitized. 

Typical schemes of a conventional and a laser induced fluorescence system are shown in Figure 2.

### 1.3. Fluorescent Compounds

The compounds that absorb visible and/or UV light are called chromophores. The compounds that emit light are called fluorophores. Compounds with fluorescent characteristics are those with several aromatic (fused) rings and/or with conjugated double bonds. There are two types of fluorophores depending on the groups: those with electron-donating groups, such as -OH, -NH_2_ and -OCH_3_ that increase fluorescence, and those with electron-withdrawing groups, such as COOH and -N=N- that reduce fluorescence. There are some exceptions, for example fluorophores such as tryptophan and tyrosine, that showed lower quantum yields than expected [12]. Compounds such as halogens, oxygen and acrylamide are also known to reduce fluorescence and will be described in the next section. Heterocyclic compounds do not fluoresce significantly unless they are attached to an aromatic ring. 

The major characteristics of a fluorophore include the fluorescence lifetime (τ_F_) and the quantum yield (Φ). Fluorescence lifetime is defined as the average time that the molecule remains in the excited state before it returns to its ground state, of the order of ns. The quantum yield is defined as the ratio of the number of the emitted photons to the number of the excited molecules. The higher the quantum yield, the higher is the fluorescence. It is almost equal to unity when the non-radiative decay rate is much smaller than the radiative (fluorescence) decay rate. 

The intensity of fluorescence (I_F_) is proportional to the molecule’s concentration in diluted solutions (ε∙b∙C < 0.05), as described from the following equation:I_F_ = k ∙ I_o_ ∙ Φ∙ (ε∙b∙C) (1)
where: 

k: a constant dependent on the instrument;

I_o_: the intensity of the incident light;

Φ: the quantum yield;

ε: the molar absorptivity;

b: the path length and

C: the molecule’s concentration.

The optical properties of standard organic fluorophores are founded in many databases and include: the wavelengths of maximum absorption and emission and their bandwidths (full width at half maximum), extinction coefficient, photoluminescence quantum yield and fluorescence lifetime [13].

### 1.4. Factors That Affect Fluorescence

The factors that affect fluorescence include solvents, temperature, pH and ionic strength as well as the presence of other substances. The decrease in fluorescence intensity caused by any factor is called quenching, while the substances that may induce this are called quenchers. The decrease in the fluorescence intensity happens either by intramolecular or by intermolecular interactions. 

One of the most frequently present quenchers is the molecular oxygen which quenches almost all known fluorophores. Other quenchers are acrylamide, amines and halogens. Table 1 summarizes examples for quenchers of some typical fluorophores. There are two mechanisms of quenching: static and dynamic. Static quenching happens when a non-fluorescent complex is formed between the fluorophore and the quencher. Dynamic quenching, otherwise called collisional quenching, happens when the quencher interferes with the fluorophore during the lifetime of the excited state. The excited molecule is then deactivated either by contacting other molecules or by intermolecular interactions (collisions). The Stern–Volmer equation describes the fluorescence quenching as follows [2]:Ι_o_/Ι = 1 + K_q_ ∙ τ_F_ ∙ [Q] (2)
where: 

I_o_: the intensity without quencher;

I: the intensity with quencher;

K_q_: the quencher rate coefficient;

τ_F_: the fluorescence lifetime and

Q: the quencher’s concentration.

**Table 1 molecules-27-04801-t001:** Examples of typical fluorophores and their quenchers.

Typical Fluorophores	Quenchers
Polycyclic aromatic hydrocarbons (PAHs)	Nitrocompounds, nitromethane [14,15]
Anthracene	Diethylaniline [16]
Tyrosine	Disulfides [17], phosphates [18]
Tryptophan, indole	Acrylamide [19,20], cations [21], anions [22]
Aromatic hydrocarbons	Aromatic and aliphatic amines, pyridinium salts [23]
Majority of known fluorophores	Oxygen [24,25]

The fluorescence of a compound can be affected from the solvent used for the fluorophore solution. The solvent effect can be observed as a shift of the spectral maxima (solvatochromic shift), as a change of the intensity of the spectral line or band and as a change of the shape and width of the band [26]. By increasing the solvent polarity, shifts of the emission spectrum to longer wavelengths (red shifts) are usually observed.

The pH parameter affects fluorescence as the structure of a molecule can be altered by altering the pH. For example, a compound can become a spherical or linear shape in different pH values [27]. The ionization of a molecule after pH modification may alter the molecular orbital of the excitable electrons. Furthermore, H^+^ ions compete with metal ions in complexation with dissolved organic matter (DOM) and thus, there are metals that increase the fluorescence in water samples [28]. 

Additionally, the effect of temperature is very important. The electrons return to the ground state by radiationless processes. By decreasing the temperature from 45 °C to 10 °C, it was found that the intensity of DOM has been increased by 48%. [29]. Finally, the ionic strength affects fluorescence by changing the conformity and by charge transfer [30].

### 1.5. Types of Laser Induced Fluorescence

Laser induced fluorescence (LIF) is classified as steady-state (or continuous wave, CW) when only the spectral information of the emission is recorded, or as time-resolved LIF, where information of the fluorescence lifetime is derived [1]. For CW LIF, where the time integrated fluorescence is recorded, both CW and pulsed laser sources can be used. In time-resolved LIF, for the fluorescence analysis in time domain, a pulsed laser is necessary with pulse duration below ns, while in frequency-domain techniques, an optical modulation of CW laser can be used. Most types of lasers can operate in both CW and pulsed mode, while there are few types that in principle cannot be run in CW mode.

The LIF spectroscopy can also be categorized according to its spectral operation, as excitation LIF when selecting the excitation wavelength or as emission LIF when the emitted light is spectrally analyzed. In excitation LIF spectroscopy, the excitation wavelength is varied using a tunable laser and each time the total emitted light is detected using a filter in front of the detector to remove any scattered laser light. In emission LIF spectroscopy, a fixed wavelength laser is used to excite the sample and the emission spectrum is measured by using a monochromator before the detector to scan the wavelength or by using a spectrometer [31].

The high intensity light, achieved from focusing the pulsed laser beam, is able to induce fluorophore excitation by the simultaneous absorption of two or more photons of lower energy (longer wavelength) [32]. Multiphoton excitation has a rather small probability, as it depends non-linearly to the light intensity (photon fluence) and becomes efficient only in the region of the focal spot. For the two-photon absorption, there is quadratic dependence on laser intensity. The excitation photons originate from one laser beam or by different laser sources. The two-photon absorption spectrum may differ in shape from the one-photon spectrum, as there are different selection rules applied for them that allow or prohibit specific transitions [33]. The two-photon absorption laser induced fluorescence (TALIF) spectroscopy could be complementary to the conventional LIF capable of revealing energy states not accessible from one-photon transition, to excites in UV range using laser of double wavelengths that are more easily available to clearly separate the excitation laser wavelength from the detected emitted light.

LIF spectroscopy is often combined with the laser-induced breakdown spectroscopy (LIBS) as an emerging analytical technique for the elemental analysis of various samples [34]. In LIBS, the sample is irradiated by a high-power pulsed laser to generate localized plasma and breakdown the material into excited ionic and atomic species, whose emission is then spectroscopically analyzed. By utilizing a second laser beam tuned to selectively excite the plasma species, a great enhancement of their emission capability is achieved [35]. 

### 1.6. Fluorescence Recording

The fluorescence signals could be recorded in several ways. In emission LIF spectroscopy, the intensity for each wavelength of the emitted light is recorded for a fixed excitation wavelength to derive the fluorescence emission spectrum. In excitation LIF spectroscopy, the intensity of the total light emitted is recorded whilst scanning the excitation wavelength to derive the fluorescence excitation spectrum. 

There are other sophisticated methods of conventional fluorescence spectroscopy that also apply in LIF, such as the synchronous fluorescence spectroscopy (SFS), the total synchronous fluorescence spectroscopy (TSFS) and the excitation-emission matrix (EEM). In SFS, the emission spectrum is recorded while both the excitation and emission wavelengths are scanned simultaneously but maintaining a constant difference (offset) between them Δλ [36]. Synchronous spectra provide more information compared to a single scan. In TSFS, the output result is a contour map that contains numerous synchronous spectra at different offsets. Finally, the EEM method that was introduced in 1977 [37,38] results in a three-dimensional (3D) plot of fluorescence excitation wavelength versus emission wavelength and intensity. It provides full detailed information that can be used to identify several fluorophores present in complex mixtures. They are easy distinguished because the maximum fluorescence intensity for each of them is resulted only from one pair of λ_ex_/λ_em_ in the matrix.

Furthermore, for quantitative measurements, the data from recording spectra need to be quantified. For this purpose, many standard fluorescence indices have been developed and are defined as the ratios of emission intensity at two different points or areas [39]. The data obtained from EEM are often reduced with statistical methods, with parallel factor analysis (PARAFAC) be the most popular discriminant analysis.

### 1.7. Interferences in Fluorescence Measurement

Scattering is a major problem encountered in fluorescence measurements. There are two types of scattering: the Raman (inelastic) and the Rayleigh (elastic) scattering. Rayleigh scattering occurs when there are molecules of smaller size than the wavelength of the excitation light, while the scattered light has the same wavelength. It is easily filtered out from the longer wavelengths of emission. The Rayleigh scatter in EEM appears as a visible diagonal line at the emission wavelength equal to the excitation wavelength, while for the second order of Rayleigh scatter, the bright line appears at the emission wavelength equal to double the excitation wavelength. Raman scatter happens because the light is absorbed and re-emitted with loss in photon energy. The loss is due to the vibrational states and the scattered light having a higher wavelength than the excitation light. Water has scattering properties from the vibration of O-H bonds, and in aqueous samples the Raman line appears in EEM as a diagonal line at excitation wavelengths from 260 to 350 nm and at emission wavelengths from 280 to 400 nm.

The inner filtering effect (IFE) is another phenomenon that affects the recorded fluorescence. The excitation IFE occurs because a part of the incident light is absorbed from the fluorophore before reaching the sample area form where the fluorescence emission is collected in. Therefore, when increasing the fluorophore concentration, the fluorescence intensity increases up to a certain point and then starts to decrease. Moreover, the emission of IFE is caused by the self-absorption of the emitted light by the fluorophore before it reaches the detector. This results in a reduction of the recorded spectrum, where the absorption overlaps with the emission which changes the shape of the recorded spectrum in the low wavelength portion. Additionally, IFE could be induced due to presence of other chromophores that absorb the excitation or the emission light. In most environmental samples, the main reason for the IFE is the naturally dissolved humic material [40]. The IFE problem is compensated by reducing the path length in the sample for the excitation or/and emission light and by diluting the water samples to have an absorbance of less than 0.1 at 254 nm [41]. 

## 2. Applications of Fluorescence in Environmental Samples

The detection of target pollutants in liquid and solid environmental samples has been extensively implemented by chromatographic methods coupled with fluorescence detectors. Fluorescence detectors are of high selectivity as there are few compounds that fluoresce. They are also of higher sensitivity (10–1000 times) compared to diode array UV detectors. A tremendous number of publications are dedicated to the determination of pollutants such as polycyclic aromatic hydrocarbons (PAHs) in water [42,43,44], in soils [45,46], in sediments [47,48], in pesticides in water [49,50], in soil [51,52], in pharmaceuticals in water [53,54], in soils [55,56], in sediments [57,58] and in metals in water [59,60] by chromatographic techniques coupled with fluorescence detectors. In all publications, a preconcentration of the samples is required, as the pollutants exist in very low concentrations. The preconcentration can be achieved by liquid–liquid extraction, solid phase extraction or solid phase microextraction for aqueous samples. For the solid samples such as soils and sediments, solid-liquid extraction and fractionation according to pH have usually been applied. Besides, fluorescence has been used for the direct characterization of quality of environmental samples without preconcentration. The applications of fluorescence in environmental samples without preconcentration are described as follows.

### 2.1. Applications of Conventional Fluorescence in Various Types of Water 

Without any sample preconcentration and/or pretreatment (except filtration if necessary), conventional fluorescence has revealed useful information about the water and wastewater quality. The majority of studies have focused on the dissolved organic matter (DOM), which is the major water constituent, while less have focused on oil pollution.

The first significant studies were those of Coble [61,62] that have characterized marine and terrestrial DOM in seawater by EEM fluorescence spectroscopy. The peaks were classified into five categories and named as A, B, C, T and M. Both peaks A and C were of humic-like fluorescence, with peak A irradiated by UV excitation and peak C by visible excitation. Peaks B and T were of tyrosine-like and tryptophan-like fluorescence, respectively. Peak M was specific for marine humic-like fluorescence.

The distinguishment between surface water from eastern (Atlantic and modified polar water) and western (Canada-basin polar water) Arctic sectors was detected by using EEM fluorescence spectroscopy [63]. In this work, the Eurasian polar water showed higher visible DOM fluorescence signals than the water from the Canada basin.

The characterization and monitoring of wastewater in surface waters has been achieved by EEM fluorescence spectroscopy [64]. The peaks T (living and dead cellular material and their exudates) and C (microbially reprocessed organic matter) from wastewater samples presented much higher intensity compared with those from natural waters. Furthermore, peak T fluorescence was highly reduced after the biological treatment process, while peak C was almost completely removed after the chlorination and reverse osmosis processes. 

Another application of EEM coupled with principal component analysis and second derivative analysis has characterized wastewater samples after each treatment process in a municipal wastewater plant [65]. Figure 3 shows an example of a wastewater sample after anaerobic treatment.

Another work [66] has used EEMs to distinguish the origin and the distribution of DOM in different water samples such as oligotrophic oceanic waters, reef waters, river waters and groundwater. The fluorophores that were identified within the samples were: humic-like A, humic-like C, marine humic-like M, tryptophan-like T1 and T2 and tyrosine-like B1 and B2. Some unknown peaks (U1 and U2) have also been identified. 

Huang [67] identified six fluorophores by EEC-PARAFAC in the eutrophicated lake Taihu during autumn. They were named A, B, C, N, M and T. The results showed that red shift happened with increasing fluorescence intensity for peaks A (UV humic-like) and C (terrestrial humic-like) while peak M had the reverse red shift. From another survey in the same lake, only four fluorophores were detected during one month [68] by EEC-PARAFAC. 

Goslan [69] studied the seasonal change of DOM and its effect on the treatment processes by EEM and by synchronous fluorescence. At least two fluorophores were identified in non-fractionated and fractionated (acids, bases, neutrals) water samples. The fractionation of samples was performed by resin according to their hydrophobicity. 

In rainwater samples [70], three fluorophores were identified by EEM: humic-like C, tyrosine-like B and tryptophan-like T. In addition, the results show that humic-like fluorescence was strongly influenced by terrestrial/anthropogenic sources, while tyrosine and tryptophane-like fluorescence was not influenced from the meteorological variables. 

Synchronized fluorescence by a conventional spectrofluorometer at Δλ = 30 nm between emission and excitation wavelengths was applied in an estuary located in northeast Brazil [71]. Four peaks were identified at wavelengths 278–280 nm (peak I), 350 nm (peak II), 385 nm (peak III) and 458–460 nm (peak IV). Peak I was due to microbial production while peaks II, III and IV were related to humic and fulvic acids.

Generally, the main fluorophores and their positions as identified by most researchers [62,72] in various types of water are summarized in Table 2.

Other researchers [73] have studied the oil dispersion in seawater by recording the emission spectra. They found that the intensity at 445 nm is indicative of higher molecular weight PAHs without having to measure the oil concentration. 

The concentrations of three PAHs were predicted by EEM-PARAFAC in urban run-offs from asphalt paved roads [74]. The PAHs that were studied were: phenanthrene, benzo (k) fluoranthene and benzo (a) pyrene.

### 2.2. Applications of LIF in Various Types of Water 

Laser induced fluorescence has also been used for the characterization of water quality without any preconcentration. The majority of these studies have used solid state lasers.

Uebel [75] applied the LIF method in order to detect water pollutants and their possible interactions with phytoplankton and break-down products (yellow-substances) in situ. They used a frequency doubled dye laser as an excitation source with a pulse energy of 10 mJ and a pulse duration of 10 ns. The substances were excited in the range from 265 nm to 400 nm while the fluorescence signal was recorded in the range of 310 nm–750 nm. Different fluorescence spectra appeared during ageing and dying of the phytoplankton.

For the detection of PAHs and oils in groundwater, Baumann [76] chose a nitrogen laser and time-resolved LIF based on the different decay times of humic substances and PAHs. They calculated the concentrations of 16 PAHs and found a good correlation with the results obtained by HPLC coupled with a fluorescence detector.

In a work of Sivaprakasam and Killinger [77], two different LIF instruments were tested for the determination of DOC and quinine sulfate in natural water samples, bottled distilled and bottled drinking water samples, where it was found that plastic-related compounds were leached into the water from the containers. One instrument utilizes a UV tunable dye laser (200–285 nm, 0.2–5 μJ, 10 Hz) with a spectrometer and CCD detector, while the other one is a portable system utilizing a fixed wavelength microchip laser (at 266 nm, 1 μJ, 8 KHz) with a gated PMT detector and bandpass interference filters. Although both systems use laser pulses of the same energy, the higher repetition rate in the latter system provides a much greater signal-to-noise ratio (SNR) measurements due to the high pulse averaging and the higher total energy output. In conjunction with the detection system, it was found to have a 10 times higher sensitivity than the first system, but with much slower processing time to obtain a full emission spectrum. It was found to have up to 100 times higher sensitivity from the best commercial portable spectrofluorometer, while it was tested in situ for the determination of plastics and DOC in seawater samples [78]. Additionally, they upgraded it to have two interchangeable microchip lasers operating at 266 nm and 355 nm, achieving the tracking of DOC in the clean ocean water by continuously operating the portable LIF system for five days [79].

The same portable LIF unit was tested to measure the fluorescence from tap water and sea water after being treated by reverse osmosis [80] and in ground and drinking water samples [81]. It was found that the deeper UV laser showed more distinct spectra with quantitative features and gave better separation of the LIF from the Raman peak allowing the detection of unique spectral features. Most of the LIF systems utilize laser sources in deep UV.

Ghervase et al. [82] have chosen the fourth harmonic output of an Nd:YAG laser for the excitation of both microbial and humic-like substances due to the high energy of the excitation photons (266 nm). They detected the impact from human and chicken waste in rivers and found that chicken waste had a specific fluorescence signature.

LIF and excitation-emission matrix (EEM) fluorescence were applied in river samples from the lower basin of the Arges River. They detected urban sewage contamination by picking up fluorescence signals, such as that from tyrosine and the presence of folded and unfolded tryptophan residues [83].

Recently, Du et al. [84] have used a UV laser at 266 nm in order to detect the presence of three aromatic amino acids in seawater (tryptophan, tyrosine, and phenylalanine) in situ. The peaks of tryptophan, tyrosine and phenylalanine were detected at 350, 300 and 280 nm, respectively, as shown in Figure 4, and their concentrations were quantified.

In a recent work [85] a small-sized spectrofluorometer operated at 278 nm was used for the identification of oil products in seawater. They found that the spectral features were changed depending on the state of the oil product.

A combination of LIBS and LIF was used in order to detect trace amounts of heavy metals in water samples. Specifically, lead was detected by using a Q-switched Nd:YAG laser to produce plasma, at 1064 nm or 532 nm [86]. In another work, lead was detected by micro-LIBS with LIF by using a 170 μJ laser pulse for ablation and a 10 μJ laser pulse for re-excitement [87]. Other metals that were detected with LIBS were cadmium after enrichment with a resin [88] and chromium [89,90]. In addition, LIBS portable compact systems were employed for in situ seawater analysis for deep sea [91,92,93].

A totally different application of LIF was used for the classification of viruses. In a recent study [94], a laser at 266 nm with a pulse repetition of 10 kHz and a power of 25 mW has been used for virological analysis of environmental samples. Although the high repetition rate of high energy pulses, the SNR was not always adequate to discriminate the virus in small concentrations. However the LIF method could significantly reduce the time and operational cost of virus analysis.

### 2.3. Applications of Conventional Fluorescence in Soils and Sediments 

Conventional fluorescence has been applied in soils and sediments after isolation of the target components such as humic substances, oils, etc.

Surface marine sediments were extracted by proper solvent and examined for the presence of bulk PAH levels by a portable fluorescence apparatus [95]. The results showed good correlation with the lab results.

EEM and synchronous scans were obtained by [96] to examine the concentrations of PAHs in soils. By using a fluorescence fingerprints library with several EEM and SFS maps for various dilutions of Romanian crude oil in methanol, they confirmed the identity of the soil pollutant. They created a calibration to estimate the pollutant concentrations.

EEM–PARAFAC was used in a study [97] conducted in lake bottom sediments of selected lobelia lakes in order to assess their properties and their origin. The optical properties of HA extracted from the sediments were compared to the parameters that describe their structural and chemical properties. Four components were identified: two protein-like (C2 and C4), one humic-like (C1), and one fulvic-like (C2). The more dominant component was C2 and the less dominant one was C4. Each of the components revealed different information. The results showed that the organic matter (OM) present in the bottom sediments from sampled lakes had autochthonic origin and consisted of mostly labile organic compounds.

Five fluorescent components were identified by EEM-PARAFAC in soils and sediments from two different estuaries in Spain and were found in both FA and HA fractions with different abundance [98]. 

Ιn surface and deep sediments from Toulon Bay in France, EEM-PARAFAC identified two components [99]: a “fresh” particulate OM in surface sediments, which produces protein like high molecular weight-DOM and low molecular weight-DOM and a “buried” particulate OM in the deeper sediment layer. In porewaters, three components were identified: C1, C2—both humic-like—and C3 of protein-like fluorescence.

From another study [100] conducted in a forested watershed, three fluorescent compounds were identified from soil/sediment HS: a terrestrial humic-like (C1), a microbial humic-like or fulvic-like (C2) and a protein-like component (C3) as shown in Figure 5. Component C3 was probably related to an autochthonous organic input to the reservoir sediments and/or phenolic compounds. They used EEM-PARAFAC spectroscopy.

EEM-PARAFAC has also distinguished soils that were treated with mineral fertilizers and organic manure from those treated with only mineral fertilizer and those without fertilization [101]. 

Synchronous fluorescence at Δλ = 20 nm has been applied in different types of soil samples and revealed the presence of five main peaks at 360, 470, 488, 502 and 512 nm. They assessed the soil humification degree in different types of soils [102]. 

### 2.4. Applications of LIF in Soils and Sediments

LIF has been applied to soil samples without any chemical and or/physical pretreatment and extraction procedures. 

The influence of a sewage-sludge addition on the OM of a Brazilian oxysol was studied by an argon laser emitted at 351 nm with an output power of 400 mW [103]. The application of LIF was performed on unfractionated soil samples, fractionated with chemical methods and fractionated by physical methods (particle size). The results indicate that every particle size fraction showed a different shape, revealing differences in organic compounds bounded to them. It was also used to obtain LIF spectra of pelletized whole soils from different origins and various depths [104]. A similar apparatus, utilizing a CW laser of 20 mW power at 405 nm for the excitation, was used for the characterization of OM in pelletized soil samples [105]. 

The LIBS spectra of soils were obtained using a Q-switched Nd:YAG laser at 1064 nm [106]. The humification degree (HD) of OM was evaluated for the first time by LIBS. The results show high correlation with those determined by LIF using a CW diode laser at 405 nm. 

The results from the previous studies were compared in a review by Senesi [107] and it was found that there is a high correlation between (HD) LIBS and (H) LIF index values, and that LIBS can be used to evaluate the humidification degree of soil organic matter (SOM).

LIF was also applied in dry and hydrated crusts by an Nd:YAG laser operating at 532 nm with 7 ns pulses [108]. The microphytobenthos (MPB) showed three peaks (two main at 570 and 650 nm, and a secondary at 720 nm) as is shown in Figure 6. Furthermore, the only slight difference between bare soil, either dry or hydrated, was the intensity of the peaks. 

## 3. Remote Sensing

LIF has the great advantage of being expended to the outdoor environment. It is a technique that can also be applied as remote sensing and, hence, it is called laser remote sensing system LIDAR (light detection and ranging). It makes it possible to analyze compounds from long distances and it can operate under full sunlight. 

There are many studies about LIDAR systems for the measurements of hydrographic parameters and substances in water. Nunes [109] has used a LIDAR using the 2nd harmonic of Nd: YAG laser system with a pulse energy of 200 mJ, a duration of 10 ns and a repetition rate of 10 Hz to measure chlorophyll a (685 nm) and DOM (from 540 to 620 nm) in deep sea water. Babichenko [110] has used a LIDAR Nd: YAG laser at 355 nm and a four-channel detector (355, 403, 450 and 680 nm) in order to measure chlorophyll a and other pigments in the subsurface water layer.

Other researchers have used the laser beam of an excimer (308 nm) and of a tunable dye laser (367 and 460 nm) to measure dissolved organic matter (DOM), chlorophyll a and other pigments in the Black Sea [111,112] and in the Danube Delta [113]. 

The disadvantages of excimer lasers are their high purchase and maintenance costs and the continuous need for gas supply compared with the solid state Nd:YAG lasers [114]. 

To the best of our knowledge, there are not any applications of LIDAR for monitoring soil quality. The only recent application had the aim to evaluate the soil surface and furrow cross-sectional area after a trailing shoe sweep [115].

## 4. Conclusions

The present study reveals the benefits of the application of fluorescence spectroscopy to environmental samples in order to assess their quality. Fluorescence can be achieved by the excitation of the compounds either by lamps or by lasers. In the first occasion, it is considered as conventional fluorescence, while in the second it is called laser induced fluorescence (LIF). Both of them have been applied mainly for the characterization of DOM in various types of water, soil and sediment samples. In a lesser extent, they have been applied for the detection of pollutants such as oil products in these samples.

In conventional fluorescence spectroscopy, the most common lamps are Xe and Hg. The most use recording technique in conventional fluorescence is EEM combined with PARAFAC analysis. It has been applied mainly for the characterization of DOM in various types of water. For the majority of the studies, the results from EEM showed the presence of peaks that have been categorized in five categories. The application of conventional spectroscopy in soil and sediment samples after pretreatment for isolation has led to the categorization of the same fluorophores.

In LIF spectroscopy, the common light source is the solid state laser Nd:YAG, as it was found to be more suitable for the detection of DOC and pollutants, followed by dye lasers that offer excitation tunability for the recording the fluorescence intensity versus emission wavelength. LIF has been applied less than conventional spectroscopy, especially in soil samples, although no sample pretreatment was necessary. Furthermore, LIF has been used for remote sensing measurements of various water systems by excimer and dye lasers.

LIF has proven to be a powerful tool for assessing the environmental quality and its use is continuously extending. Lasers have the advantages of high sensitivity and selectivity, as well as being versatile sources. Furthermore, the problem of quenching is minimized because laser sources have very high intensities compared with the potential quenching. The disadvantage is the restriction in excitation tunability that makes it difficult to record the EEM in order to gain more information about the complexity of the environmental samples.

## Figures and Tables

**Figure 1 molecules-27-04801-f001:**
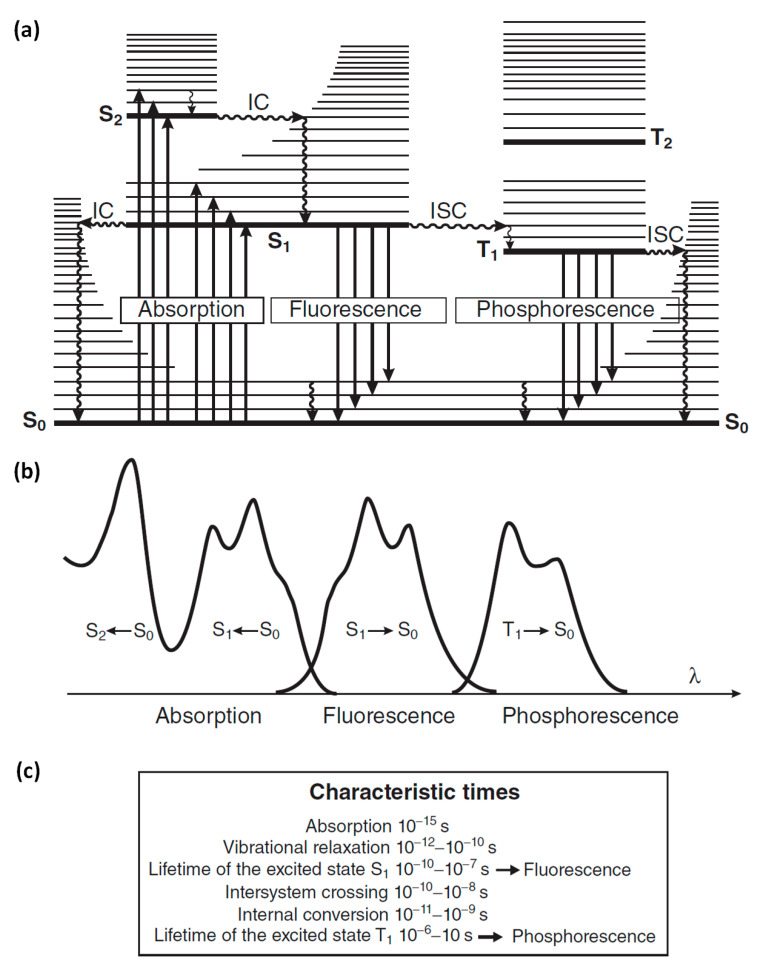
(**a**) Perrin–Jablonski diagram with the possible radiative transitions (straight arrows) of absorption, fluorescence and phosphorescence, as well as the non-radiative transitions (wavy arrows) of vibrational relaxation, internal conversion (IC) and intersystem crossing (ISC). (**b**) Illustration of the spectra for the radiative transitions between electronic states that shown in (**a**). (**c**) Characteristic times for each transition. Reprinted from [1]. 2012, John Wiley and Sons.

**Figure 2 molecules-27-04801-f002:**
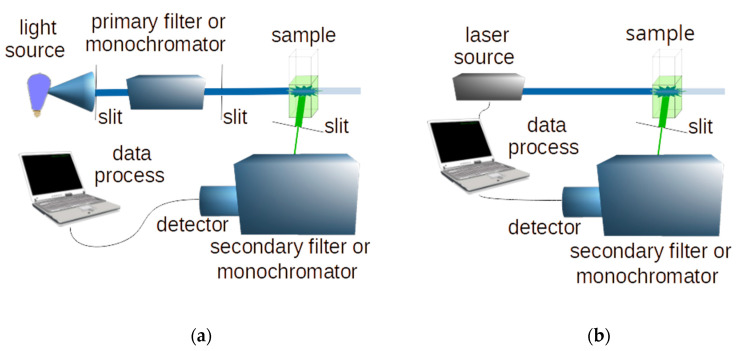
Schemes of a conventional (**a**) and a laser-induced (**b**) fluorescence spectroscopy system.

**Figure 3 molecules-27-04801-f003:**
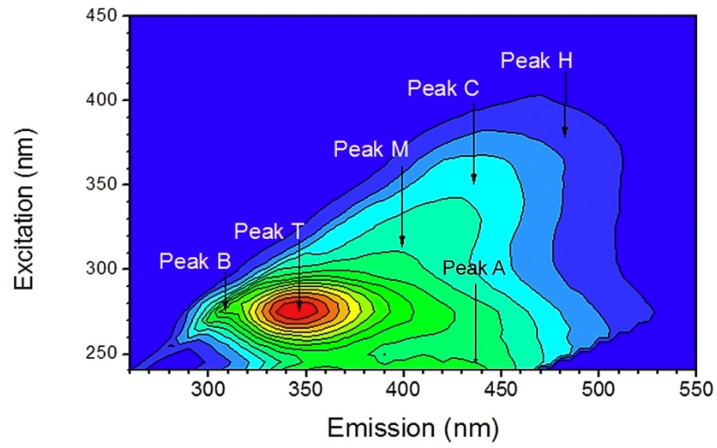
EEM feature and the position of peaks A, C, B, M, H and T in a wastewater sample after anaerobic treatment. Adapted with permission from Ref. [65]. 2018, Elesevier.

**Figure 4 molecules-27-04801-f004:**
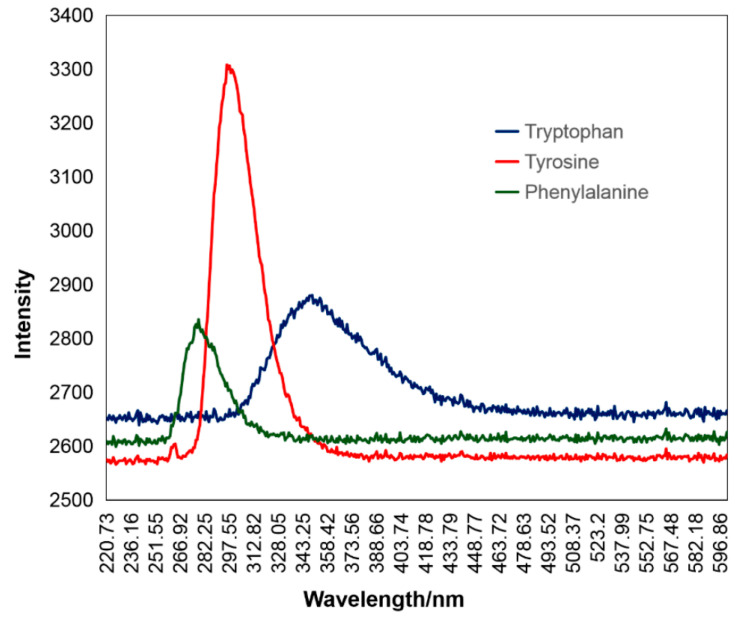
LIF spectra of the three aromatic amino acids as presented by [84].

**Figure 5 molecules-27-04801-f005:**
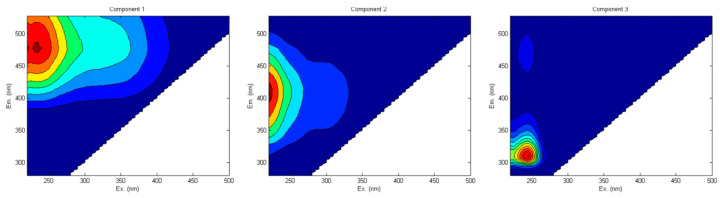
The three fluorescent components identified by EEM-PARAFAC. Adapted with permission from Ref. [100]. 2017, Springer Nature.

**Figure 6 molecules-27-04801-f006:**
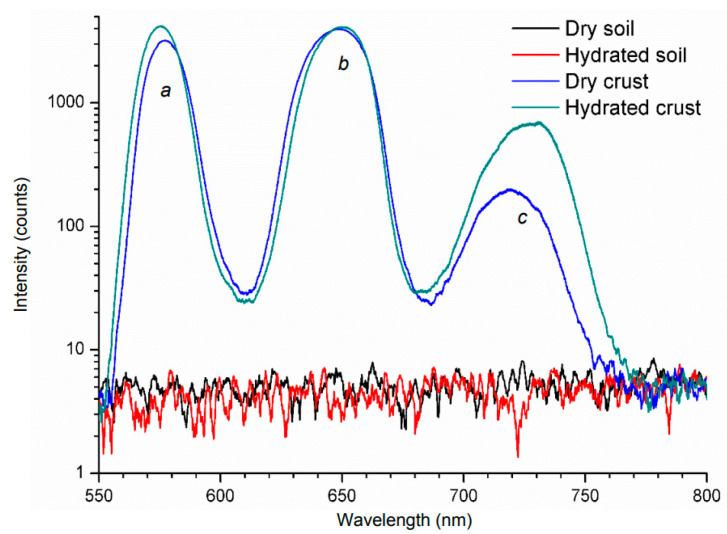
LIF spectra obtained from soils and crust samples [108].

**Table 2 molecules-27-04801-t002:** Main fluorophores and their positions.

Name	Letter	Ex (nm)/Em (nm)
Humic-like	C	340–360/420–480
Fulvic-like	A	240–260/380–460
Tyrosine-like	B (B1, B2)	265–285/290–310
Tryptophan-like	T (T1, T2)	265–285/290–340
Microbial-like (Marine-like)	M	310–330/390–410
Humic-like	H	370–390/480–500

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
