# Peer review of "Review of Fluorescence Spectroscopy in Environmental Quality Applications"

_molecules, 2022, doi:10.3390/molecules27154801_

Round 1

Reviewer 1 Report

The presented review briefly, but quite fully for such a volume, describes the use of fluorescence for the analysis of environmental objects. There are a few small comments that can be eliminated, after which the article can be accepted for publication.

1.The references should be checked. Is the numeration correct? For example, line 25 – reference in the brackets (Lakowizc 2006) does not correspond to reference rules for the journal. Please, check.

2. Section 1 (Introduction) contains very small amount of references despite on many theoretical information. It should be revised.

3. It is better to add the table to the section 1.3 with the list of fluorescent compounds used with the corresponding references. This table may be improved with the list of quenchers described by now (with the examples from articles). It is recommendation for better perception and reading.

Author Response

  1. The references should be checked. Is the numeration correct? For example, line 25 – reference in the brackets (Lakowizc 2006) does not correspond to reference rules for the journal. Please, check.

All the references were checked. Some of them have been replaced and/or repositioned. The reference in line 25 has been removed.

  1. Section 1 (Introduction) contains very small amount of references despite on many theoretical information. It should be revised.

In the first section, 21 citations have been added and 1 more in the second section of the text. Totally, the manuscript has been enriched with 22 more citations.

  1. It is better to add the table to the section 1.3 with the list of fluorescent compounds used with the corresponding references. This table may be improved with the list of quenchers described by now (with the examples from articles). It is recommendation for better perception and reading.

A table has been added in the section 1.3 as indicated (Table 1). The table contains a list of quenchers and their citations.

Round 2

Reviewer 1 Report

In my opinion, this manuscript is improved according to the notes of reviewers and can be accepted 

Author Response

Thanks very much for your precious comments

Reviewer 2 Report

I am satisfied with all the modifications, so that I think this version can be suited for the publication.

I found small points that should be modified before the publication.

1. line 246, "that apply also applied" 

2. line 294, "fluorescent" should be "fluoresce" or "are fluorescent"

3. line 437-438, the sentence is strange.

4. line 483, explanation of OM is missing. (later, in line493, OM is explained)

5.  line 527, "The present study review"

6. line 549, "Futhermore Furthermore"  

Author Response

Dear Reviewer 2,

we have done the minor corrections and  few others that we found in order to improve the text

Thanks for the precious suggestions